# MedMentions: A Large Biomedical Corpus Annotated with UMLS Concepts

**Sunil Mohan**                                                    SMOHAN@CHANZUCKERBERG.COM

**Donghui Li**                                                    DLI@CHANZUCKERBERG.COM
*Chan Zuckerberg Initiative,*
*Redwood City, CA 94063 USA*

## Abstract

This paper presents the formal release of MedMentions, a new manually annotated resource for the recognition of biomedical concepts. What distinguishes MedMentions from other annotated biomedical corpora is its size (over 4,000 abstracts and over 350,000 linked mentions), as well as the size of the concept ontology (over 3 million concepts from UMLS 2017) and its broad coverage of biomedical disciplines. In addition to the full corpus, a sub-corpus of MedMentions is also presented, comprising annotations for a subset of UMLS 2017 targeted towards document retrieval. To encourage research in Biomedical Named Entity Recognition and Linking, data splits for training and testing are included in the release, and a baseline model and its metrics for entity linking are also described.

## 1. Introduction

One recognized challenge in developing automated biomedical entity extraction systems is the lack of richly annotated training datasets. While there are a few such datasets available, the annotated corpus often contains no more than a few thousand annotated entity mentions. Additionally, the annotated entities are limited to a few types of biomedical concepts such as diseases [Doğan and Lu, 2012], gene ontology terms [Van Auken et al., 2014], or chemicals and diseases [Li et al., 2016]. Researchers targeting the recognition of multiple biomedical entity types have had to resort to specialized machine learning techniques for combining datasets labelled with subsets of the full target set, e.g. using multi-task learning [Crichton et al., 2017], or a modified Conditional Random Field cost which allows un-labeled tokens to take any labels not in the current dataset's target set [Greenberg et al., 2018]. To promote the development of state-of-the-art entity linkers targeting a more comprehensive coverage of biomedical concepts, we decided to create a large concept-mention annotated gold standard dataset named 'MedMentions' [Murty et al., 2018].

With the release of MedMentions, we hope to address two key needs for developing better biomedical concept recognition systems: (i) a much broader coverage of the fields of biology and medicine through the use of the Unified Medical Language System (UMLS) as the target ontology, and (ii) a significantly larger annotated corpus than available today, to meet the data demands of today's more complex machine learning models for concept recognition.

The paper begins with an introduction to the MedMentions annotated corpus, including a sub-corpus aimed at information retrieval systems. This is followed by a comparison with a few other large datasets annotated with biomedical entities. Finally, to promote further research on large

ontology named entity recognition and linking, we present metrics for a baseline end-to-end concept recognition (entity type recognition and entity linking) model trained on the MedMentions corpus.

## 2. Introducing MedMentions

### 2.1 The Documents

We randomly selected 5,000 abstracts released in PubMed®[1] between January 2016 and January 2017. Upon review, some abstracts were found to be outside the biomedical fields or not written in English. These were discarded, leaving a total of 4,392 abstracts in the corpus.

### 2.2 Concepts in UMLS

The Metathesaurus of UMLS [Bodenreider, 2004] combines concepts from over 200 source ontologies. It is therefore the largest single ontology of biomedical concepts, and was a natural choice for constructing an annotated resource with broad coverage in biomedical science.

In this paper, we will use *entities* and *concepts* interchangeably, to refer to UMLS concepts. The 2017 AA release of the UMLS Metathesaurus contains approximately 3.2 million unique concepts. Each concept has a unique id (a "CUID") and primary name and a set of aliases, and is linked to all the source ontologies it was mapped from. Each concept is also linked to one or more Semantic Types – the UMLS guidelines are to link each concept to the most specific type(s) available. Each Semantic Type also has a unique identifier ("TUI") and a name. The Metathesaurus contains 127 Semantic Types, arranged in a "is-a" hierarchy. About 91.7% of the concepts are linked to exactly one semantic type, approximately 8% to two types, and a very small number to more than two types.

### 2.3 Annotating Concept Mentions

We recruited a team of professional annotators with rich experience in biomedical content curation to exhaustively annotate UMLS entity mentions from the abstracts.

The annotators used the text processing tool GATE[2] (version 8.2) to facilitate the curation. All the relevant scientific terms from each abstract were manually searched in the 2017 AA (full) version of the UMLS metathesaurus[3] and the best matching concept was retrieved. The annotators were asked to annotate the most specific concept for each mention, without any overlaps in mentions.

To gain insight on the annotation quality of MedMentions, we randomly selected eight abstracts from the annotated corpus. Two biologists ("Reviewers") who did not participate in the annotation task then each reviewed four abstracts and the corresponding concepts in MedMentions. The abstracts contained a total of 469 concepts. Of these 469 concepts, the agreement between Reviewers and Annotators was 97.3%, estimating the *precision* of the annotation in MedMentions. Due to the size of UMLS, we reasoned that no human curators would have knowledge of the entire UMLS, so we did not perform an evaluation on the recall. We are working on getting more detailed IAA (Inter-annotator agreement) data, which will be released when that task is completed.

---

1. http://pubmed.gov
2. https://gate.ac.uk/
3. http://umlsks.nlm.nih.gov

## 2.4 MedMentions ST21pv

Entity linking / labeling methods have prominently been used as the first step towards relationship extraction, e.g. the BioCreative V CDR task for Chemical-Disease relationship extraction [Li et al., 2016], and for indexing for entity-based document retrieval, e.g. as described in the BioASQ Task A for semantic indexing [Nentidis et al., 2018]. One of our goals in building a more comprehensive annotated corpus was to provide indexing models with a larger ontology than MeSH (used in BioASQ Task A and PubMed) for semantic indexing, to support more specific document retrieval queries from researchers in all biomedical disciplines.

UMLS does indeed provide a much larger ontology (see Table 6). However UMLS also contains many concepts that are not as useful for specialized document retrieval, either because they are too broad so not discriminating enough (e.g. *Groups [cuid = C0441833]*, *Risk [C0035647]*), or cover peripheral and supplementary topics not likely to be used by a biomedical researcher in a query (e.g. *Rural Area [C0178837]*, *No difference [C3842396]*).

Filtering UMLS to a subset most useful for semantic indexing is going to be an area of ongoing study, and will have different answers for different user communities. Furthermore, targeting different subsets will also impact machine learning systems designed to recognize concepts in text. As a first step, we propose the "*ST21pv*" subset of UMLS, and the corresponding annotated sub-corpus *MedMentions ST21pv*. Here "ST21pv" is an acronym for "21 Semantic Types from Preferred Vocabularies", and the ST21pv subset of UMLS was constructed as follows:

1. We eliminated all concepts that were only linked to semantic types at levels 1 or 2 in the UMLS Semantic Type hierarchy with the intuition that these concepts would be too broad. We also limited the concepts to those in the *Active* subset of the 2017 AA release of UMLS.

2. We then selected 21 semantic types at levels 3–5 based on biomedical relevance, and whether MedMentions contained sufficient annotated examples. Only concepts *mapping into* one of these 21 types (i.e. *linked* to one of these types or to a descendant in the type hierarchy) were considered for inclusion. As an example, the semantic type *Archaeon [T194]* was excluded because MedMentions contains only 25 mentions for 15 of the 5,418 concepts that map into this type (Table 2).

   Since our primary purpose for ST21pv is to use annotations from this subset as an aid for biomedical researchers to retrieve relevant papers, some types were eliminated if most of their member concepts were considered by our staff biologists as not useful for this task. An example is *Qualitative Concept [T080]*, which contains frequently mentioned concepts like *Associated with [C0332281]*, *Levels [C0441889]* and *High [C0205250]*.

3. Finally, we selected 18 'prefered' source vocabularies (Table 1), and excluded any concepts that were not linked in UMLS to at least one of these sources. These vocabularies were selected based on usage and relevance to biomedical research[4], with an emphasis on gene function, disease and phenotype, structure and anatomy, and drug and chemical entities.

Table 2 gives a detailed breakdown of a portion of the semantic type hierarchy in UMLS 2017 AA Active. The rows in bold are the 21 types in ST21pv, and any descendants of these types have

---

4. An example of ontology usage or popularity can be found in the "Ontology Visits" statistics available at https://bioportal.bioontology.org.

| Ontology Abbrev. | Name |
|---|---|
| CPT | Current Procedural Terminology |
| FMA | Foundational Model of Anatomy |
| GO | Gene Ontology |
| HGNC | HUGO Gene Nomenclature Committee |
| HPO | Human Phenotype Ontology |
| ICD10 | International Classification of Diseases, Tenth Revision |
| ICD10CM | ICD10 Clinical Modification |
| ICD9CM | ICD9 Clinical Modification |
| MDR | Medical Dictionary for Regulatory Activities |
| MSH | Medical Subject Headings |
| MTH | UMLS Metathesaurus Names |
| NCBI | National Center for Biotechnology Information Taxonomy |
| NCI | National Cancer Institute Thesaurus |
| NDDF | First DataBank MedKnowledge |
| NDFRT | National Drug File – Reference Terminology |
| OMIM | Online Mendelian Inheritance in Man |
| RXNORM | NLM's Nomenclature for Clinical Drugs for Humans |
| SNOMEDCT_US | US edn. of the Systematized Nomenclature of Medicine-Clinical Terms |

Table 1: The restricted set of source ontologies for MedMentions ST21pv.

been pruned and their counts rolled up. The counts therefore are for concepts *linked* to the corresponding type for the non-bold rows, and *mapped* to the ST21pv types for the rows in bold. Note that some concepts in UMLS are linked to multiple semantic types. The prefix *MM-* in the column name indicates the counts are for concepts mentioned in MedMentions. The full MedMentions corpus contains 2,473 mentions of 685 concepts that are not members of the 2017 AA Active release. These were eliminated as part of step 1. The other non-bold rows in the table represent semantic types excluded in steps 1 and 2, corresponding to a total of 135,986 mentions of 6,002 unique concepts. A further 10,755 mentions of 2,618 concepts were eliminated in step 3. As a result of all this filtering, the target ontology for MedMentions ST21pv (MM-ST21pv) contains 2,327,250 concepts and 203,282 concept mentions.

Examples of broad concepts eliminated by selecting semantic types at level 3 or higher:

- C1707689: "Design", linked to T052: Activity, level=2

- C0029235: "Organism" linked to T001: Organism, level=3

- C0520510: "Materials" linked to T167: Substance, level=3

## 2.5 MedMentions Corpus Statistics

The MedMentions corpus consists of 4,392 abstracts randomly selected from those released on PubMed between January 2016 and January 2017. Table 3 shows some descriptive statistics for

| TypeName | TypeID | Level | nConcepts | MM-nConcepts | MM-nDocs | MM-nMentions |
|---|---|---|---|---|---|---|
| Event | T051 | 1 | 185 | 16 | 146 | 292 |
| Activity | T052 | 2 | 420 | 152 | 2,615 | 7,253 |
| Behavior | T053 | 3 | 84 | 23 | 191 | 447 |
| Social Behavior | T054 | 4 | 924 | 171 | 382 | 982 |
| Individual Behavior | T055 | 4 | 857 | 149 | 352 | 1,012 |
| Daily or Recreational Activity | T056 | 3 | 808 | 71 | 218 | 863 |
| Occupational Activity | T057 | 3 | 739 | 130 | 465 | 891 |
| **Health Care Activity** | **T058** | **4** | **390,903** | **3,760** | **3,593** | **26,300** |
| **Research Activity** | **T062** | **4** | **1,598** | **538** | **3,166** | **9,965** |
| Governmental or Regulatory Activity | T064 | 4 | 516 | 61 | 94 | 188 |
| Educational Activity | T065 | 4 | 2,241 | 74 | 172 | 554 |
| Machine Activity | T066 | 3 | 155 | 37 | 125 | 288 |
| Phenomenon or Process | T067 | 2 | 1,615 | 154 | 900 | 2,034 |
| **Injury or Poisoning** | **T037** | **3** | **104,583** | **274** | **521** | **1,895** |
| Human-caused Phenomenon or Process | T068 | 3 | 560 | 48 | 173 | 295 |
| Environmental Effect of Humans | T069 | 4 | 68 | 27 | 62 | 190 |
| Natural Phenomenon or Process | T070 | 3 | 749 | 306 | 956 | 2,831 |
| **Biologic Function** | **T038** | **4** | **233,423** | **5,587** | **3,955** | **43,514** |
| Entity | T071 | 1 | 23 | 6 | 81 | 109 |
| Physical Object | T072 | 2 | 42 | 6 | 29 | 79 |
| Organism | T001 | 3 | 118 | 41 | 377 | 1,038 |
| **Virus** | **T005** | **4** | **18,128** | **131** | **174** | **1,105** |
| **Bacterium** | **T007** | **4** | **350,363** | **376** | **325** | **2,051** |
| Archaeon | T194 | 4 | 5,428 | 13 | 8 | 25 |
| **Eukaryote** | **T204** | **4** | **806,577** | **1,243** | **1,428** | **8,640** |
| **Anatomical Structure** | **T017** | **3** | **196,416** | **2,972** | **2,538** | **20,778** |
| Manufactured Object | T073 | 3 | 6,152 | 455 | 1,156 | 3,615 |
| **Medical Device** | **T074** | **4** | **58,801** | **468** | **565** | **2,406** |
| Research Device | T075 | 4 | 119 | 19 | 192 | 365 |
| Clinical Drug | T200 | 4 | 129,570 | 27 | 22 | 61 |
| Substance | T167 | 3 | 9,036 | 98 | 676 | 1,769 |
| **Body Substance** | **T031** | **4** | **2,055** | **108** | **475** | **1,258** |
| **Chemical** | **T103** | **4** | **435,397** | **5,614** | **2,734** | **38,225** |
| **Food** | **T168** | **4** | **7,041** | **174** | **286** | **1,462** |
| Conceptual Entity | T077 | 2 | 758 | 160 | 1,470 | 2,997 |
| Organism Attribute | T032 | 3 | 678 | 133 | 1,405 | 3,732 |
| **Clinical Attribute** | **T201** | **4** | **85,018** | **271** | **858** | **2,027** |
| **Finding** | **T033** | **3** | **308,234** | **3,143** | **3,577** | **18,435** |
| Idea or Concept | T078 | 3 | 3,541 | 389 | 2,839 | 9,348 |
| Temporal Concept | T079 | 4 | 3,742 | 431 | 2,621 | 10,169 |
| Qualitative Concept | T080 | 4 | 4,249 | 1,037 | 4,122 | 31,485 |
| Quantitative Concept | T081 | 4 | 9,106 | 904 | 3,441 | 19,995 |
| **Spatial Concept** | **T082** | **4** | **42,799** | **1,318** | **2,992** | **13,386** |
| Functional Concept | T169 | 4 | 3,549 | 721 | 3,979 | 23,661 |
| **Body System** | **T022** | **5** | **570** | **60** | **257** | **517** |
| Occupation or Discipline | T090 | 3 | 529 | 114 | 321 | 565 |
| **Biomedical Occupation or Discipline** | **T091** | **4** | **1,107** | **191** | **484** | **938** |
| **Organization** | **T092** | **3** | **2,695** | **291** | **882** | **2,255** |
| Group | T096 | 3 | 53 | 22 | 479 | 1,046 |
| **Professional or Occupational Group** | **T097** | **4** | **5,704** | **261** | **623** | **1,856** |
| **Population Group** | **T098** | **4** | **2,556** | **244** | **1,644** | **6,319** |
| Family Group | T099 | 4 | 372 | 56 | 233 | 816 |
| Age Group | T100 | 4 | 120 | 43 | 628 | 2,157 |
| Patient or Disabled Group | T101 | 4 | 259 | 37 | 1,520 | 6,300 |
| Group Attribute | T102 | 3 | 130 | 24 | 94 | 154 |
| **Intellectual Product** | **T170** | **3** | **30,864** | **1,110** | **2,660** | **11,375** |
| Language | T171 | 3 | 1,063 | 15 | 39 | 99 |
| Not in 2017 AA Active | - | - | - | 685 | 1,088 | 2,473 |

Table 2: UMLS semantic type hierarchy pruned at the 21 types in *ST21pv* (in bold), showing number of concepts and mentions in MedMentions. Counts in non-bold rows are for concepts *linked* to the corresponding type, and for the ST21pv types (bold) concepts *mapped* to those types.

| | MedMentions | MM-ST21pv |
|---|---:|---:|
| Total nbr. documents | 4,392 | 4,392 |
| Total nbr. unique concepts mentioned | 34,724 | 25,419 |
| Total nbr. mentions | 352,496 | 203,282 |
| Avg nbr mentions / doc | 80.3 | 46.3 |
| Total nbr. sentences | 42,602 | 42,602 |
| Total nbr. tokens | 1,176,058 | 1,176,058 |
| Total nbr. tokens annotated | 579,839 | 366,742 |
| Proportion of tokens annotated | 49.3% | 31.2% |
| Avg nbr. tokens / mention | 1.6 | 1.8 |
| Avg nbr. tokens / doc | 267.8 | 267.8 |
| Avg nbr. annotated tokens / doc | 132.0 | 83.5 |
| Avg nbr. sentences / doc | 9.7 | 9.7 |
| Avg nbr. tokens / sentence | 27.6 | 27.6 |

Table 3: Some statistics describing MedMentions. Sentence-splitting and tokenization was done using Stanford CoreNLP [Manning et al., 2014] ver. 3.8 and its Penn TreeBank tokenizer.

the MedMentions corpus and its ST21pv subset. The tokenization and sentence splitting were performed using Stanford CoreNLP[5] [Manning et al., 2014].

Due to the size of UMLS, only about 1% of its concepts are covered in MedMentions. So a major part of the challenge for machine learning systems trained to recognize these concepts is 'unseen labels' (often called "zero-shot learning", e.g. [Palatucci et al., 2009, Srivastava et al., 2018, Xian et al., 2017]). As part of the release, we also include a 60% - 20% - 20% random partitioning of the corpus into training, development (often called 'validation') and test subsets. These are described in Table 4. As can be seen from the table, about 42% of the concepts in the test data do not occur in the training data, and 38% do not occur in either training or development subsets.

### 2.6 Accessing MedMentions

The MedMentions resource has been published at `https://github.com/chanzuckerberg/MedMentions`. The corpus itself is in PubTator [Wei et al., 2013] format, which is described on the release site. The corpus consists of PubMed abstracts, each identified with a unique PubMed identifier (PMID). Each PubMed abstract has Title and Abstract texts, and a series of annotations of concept mentions. Each concept mention identifies the portion of the document text comprising the mention, and the UMLS concept. A separate file for the ST21pv sub-corpus is also included in the release.

The release also includes three lists of PMID's that partition the corpus into a 60% - 20% - 20% split defining the Training, Development and Test subsets. Researchers are encouraged to train their models using the Training and Development portions of the corpus, and publish test results on the held-out Test subset of the corpus.

---

5. `https://stanfordnlp.github.io/CoreNLP/`

|                                                 | Training | Dev    | Test   |
| ----------------------------------------------- | -------- | ------ | ------ |
| Nbr. documents                                  | 2,635    | 878    | 879    |
| Nbr. mentions                                   | 122,241  | 40,884 | 40,157 |
| Nbr. unique concepts mentioned                  | 18,520   | 8,643  | 8,457  |
| Nbr. concepts overlapping with Training         |          | 4,984  | 4,867  |
| Proportion of concepts overlapping with Training |         | 57.7%  | 57.5%  |
| Nbr. concepts overlapping with Training + Dev   |          |        | 5,217  |
| Proportion of concepts overlapping with Training + Dev |   |        | 61.7%  |

Table 4: The Training-Development-Test splits for *MM-ST21pv* are a random 60% - 20% - 20% partition.

|                                | GENIA         | ITI TXM            | CRAFT v1.0       | MedMentions |
| ------------------------------ | ------------- | ------------------ | ---------------- | ----------- |
| Nbr. Documents                 | 2,000         | (full text) 455    | (full text) 67   | 4,392       |
| Nbr. Sentences                 | $\sim$ 21k    | $\sim$ 94k         | $\sim$ 21k       | 42,602      |
| Nbr. Tokens (PTB)              | $\sim$ 440,000 | $\sim$ 2.7M       | $\sim$ 560,000   | 1,176,058   |
| Nbr. Tokens Annotated          | n/a           | n/a                | n/a              | 579,839     |
| Nbr. Mentions                  | $\sim$ 100,000 | $\sim$ 324k       | 99,907           | 352,496     |
| Nbr. unique Concepts mentioned | 36            | n/a                | 4,319            | 34,724      |
| Ontology Nbr. Concepts         | 36            | n/a                | 862,763          | 3,271,124   |

Table 5: Comparing the GENIA, IIT TXM, CRAFT and MedMentions corpora. Notes: (1) Documents in the GENIA and MedMentions corpora are abstracts. (2) The counts for ITI TXM are estimates. Some documents were annotated by multiple curators, and left as separate versions in the corpus.

## 3. A Comparison With Some Related Corpora

There have been several gold standard (manually annotated) corpora of biomedical scientific literature made publicly available. Some of the larger ones are described below.

**GENIA:** [Ohta et al., 2002, Kim et al., 2003] One of the earliest 'large' biomedical annotated corpora, it is aimed at biomedical Named Entity Recognition, where the annotations are for 36 biomedical Entity Types. The dataset consists of 2,000 MEDLINE abstracts about "biological reactions concerning transcription factors in human blood cells", collected by searching on MEDLINE using the MeSH terms *human*, *blood cells* and *transcription factors*. An extended version (2,404 abstracts), with a smaller ontology (six types) was later used for the JNLPBA 2004 NER task [Kim et al., 2004].

**ITI TXM Corpora:** [Alex et al., 2008] Among the largest gold standard biomedical annotated corpora previously available, this consists of two sets of full-length papers obtained from

PubMed and PubMed Central: 217 articles focusing on protein-protein interactions (PPI) and 238 articles on tissue expressions (TES). The PPI and TES corpora were annotated with entities from NCBI Taxonomy, NCBI Reference Sequence Database, and Entrez Gene. The TES corpus was also annotated with entities from Chemical Entities of Biological Interest (ChEBI) and Medical Subject Headings (MeSH). The concepts were grouped into 15 entity types, and these type labels were included in the annotations. In addition to concept mentions, the corpus also includes relations between entities.

The statistics (Table 5) for this corpus [Alex et al., 2008] are a little confusing, since not all sections of the articles were annotated. Furthermore some articles were annotated by more than one biologist, and each annotated version was incorporated into the corpus as a separate document.

**CRAFT:** [Bada et al., 2012] The Colorado Richly Annotated Full-Text (CRAFT) Corpus is another large gold standard corpus annotated with a diverse set of biomedical concepts. It consists of 67 full-text open-access biomedical journal articles, downloaded from PubMed Central, covering a wide range of disciplines, including genetics, biochemistry and molecular biology, cell biology, developmental biology, and computational biology. The text is annotated with concepts from 9 biomedical ontologies: ChEBI, Cell Ontology, Entrez Gene, Gene Ontology (GO) Biological Process, GO Cellular Component, GO Molecular Function, NCBI Taxonomy, Protein Ontology, and Sequence Ontology. The latest release of CRAFT[6] reorganizes this into ten Open Biomedical Ontologies. The corpus also contains exhaustive syntactic annotations. Table 5 gives a comparison of the sizes of CRAFT against the other corpora mentioned here.

MedMentions can be viewed as a supplement to the CRAFT corpus, but with a broader coverage of biomedical research (over four thousand abstracts compared to the 67 articles in CRAFT). Through the larger set of ontologies included within UMLS, MedMentions also contains more comprehensive annotation of concepts from some biomedical fields, e.g. diseases and drugs (see Table 1 for a partial list of the ontologies included in UMLS).

**BioASQ Task A:** [Nentidis et al., 2018] The Large Scale Semantic Indexing task considers assigning MeSH headings for 'important' concepts to each document. The training data is very large, but with a smaller target concept vocabulary (see Table 6), and annotation (by NCBI) is at the document level rather than at the mention level.

**Relation / Event Extraction Corpora:** Most recently developed manually annotated datasets of biomedical scientific literature have focused on the task of extracting biomedical events or relations between entities. These datasets have been used for shared tasks in biomedical NLP workshops like BioCreative, e.g. BC5-CDR [Li et al., 2016] which focuses on Chemical-Disease relations, and BioNLP, e.g. the BioNLP 2013 Cancer Genetics (CG) and Pathway Curation tasks [Pyysalo et al., 2015] where the main goal is to identify events involving entities. While these datasets include entity mention annotations, they are typically focused on a small set of entity types, and the sizes of the corpora are also smaller (1,500 document abstracts in BC5-CDR, 600 abstracts in CG, and 525 abstracts in PC). Machine learning mod-

---

6. CRAFT ver. 3.0, https://github.com/UCDenver-ccp/CRAFT

|                                          | BioASQ Task A (2018) | MedMentions ST21pv |
|------------------------------------------|---------------------:|-------------------:|
| Nbr. Training+Dev Documents              | 13.48M               | 3,513              |
| Average nbr. unique Concepts / Document  | 12.7                 | 22.4               |
| Total nbr. Concepts in Target Ontology   | 28,956               | 2,327,250          |
| Ontology Coverage in Training+Dev Data   | 98%                  | 1.1%               |
| Concept overlap, Test v/s Training+Dev   | (est.) $\sim 98\%$   | 61.7%              |

Table 6:   Comparing BioAsq Task A (2018) with MedMentions ST21pv. About 38% of the concepts mentioned in MedMentions ST21pv Test data have no mentions in the Training or Development subsets.

els for relation extraction take as input a text annotated with entity mentions, so recognizing biomedical concepts remains an important foundational task.

## 4. Concept Recognition with MedMentions ST21pv

Our main goal in constructing and releasing MedMentions is to promote the development of models for recognizing biomedical concepts mentioned in scientific literature. To help jumpstart this research, we now present a baseline modeling approach trained using the Training and Development splits of MedMentions ST21pv, and its metrics on the MM-ST21pv Test set. A subset of a pre-release version of MedMentions was also used by [Murty et al., 2018] to test their hierarchical entity linking model.

### 4.1 A Brief Note on Concept Recognition Metrics

We measure the performance of the model described below at both the *mention level* (also referred to as phrase level) and the *document level*. Concept annotations in MedMentions identify an exact span of text using start and end positions, and annotate that span with an entity type identifier and entity identifier. Concept recognition models like the one described below will output predictions in a similar format. The performance of such models is usually measured using *mention level precision*, *recall* and *F1 score* as described in [Sang and Meulder, 2003]. Here we are interested in measuring the entity resolution performance of the model: a prediction is counted as a true positive ($tp$) only when the predicted text span as well as the linked entity (and by implication the entity type) matches with the gold standard reference. All other predicted mentions are counted as false-positives ($fp$), and all un-matched reference entity mentions as false-negatives ($fn$). These counts are used to compute the following metrics:

$$precision = tp/(tp + fp)$$
$$recall = tp/(tp + fn)$$
$$F_1\ score = \left(\frac{precision^{-1} + recall^{-1}}{2}\right)^{-1} = 2 \cdot \frac{precision \cdot recall}{precision + recall}$$

| Metric | Mention-level | Document-level |
|--------|--------------|----------------|
| Precision | 0.471 | 0.536 |
| Recall | 0.436 | 0.561 |
| F1 Score | 0.453 | 0.548 |

Table 7: Entity linking metrics for the TaggerOne model on MedMentions ST21pv.

Mention level metrics would be the primary concept recognition metrics of interest when, for example, the model is used as a component in a relation extraction system. As another example, the Disease recognition task in BC5-CDR described above uses mention level metrics.

Document level metrics are computed in a similar manner, after mapping all concept mentions as entity labels directly to the document, and discarding all associations with spans of text that identify the locations of the mentions in the document. For example, a document may contain three mentions of the concept *Breast Carcinoma* in three different parts (spans) of the document text; for document level metrics they are all mapped to one label on the document. Document level metrics are useful in information retrieval when the goal is simply to retrieve the entire matching document, and are used in the BioASQ Large Scale Semantic Indexing task mentioned earlier.

## 4.2 End-to-end Entity Recognition and Linking with TaggerOne

TaggerOne [Leaman and Lu, 2016] is a semi-Markov model doing joint entity type recognition and entity linking, with perceptron-style parameter estimation. It is a flexible package that handles simultaneous recognition of multiple entity types, with published results near state-of-the-art, e.g. for joint Chemical and Disease recognition on the BC5-CDR corpus. We used the package without any changes to its modeling features.

The MM-ST21pv data presented to TaggerOne was modified as follows: for each mention of a concept in the data, the Semantic Type label was modified to one of the 21 semantic types (Table 2) that concept mapped into. Thus each mention was labeled with one of 21 entity types, as well as linked to a specific concept from the ST21pv subset of UMLS. Twenty one lexicons of primary and alias names for each concept in the 21 types, extracted from UMLS 2017 AA Active, were also provided to TaggerOne.

Training was performed on the Training split, and the Development split was provided as holdout data (validation data, used for stopping training). The model was trained with the parameters: `REGULARIZATION = 0`, `MAX_STEP_SIZE = 1.5`, for a maximum of 10 epochs with patience (`iterationsPastLastImprovement`) of 1 epoch. Our model took 9 days to train on a machine equipped with Intel Xeon Broadwell processors and over 900GB of RAM.

When TaggerOne detects a concept in a document it identifies a span of text (start and end positions) within the document, and labels it with an entity type and links it to a concept from that type. Metrics are calculated by comparing these concept predictions against the reference or ground truth (i.e. the annotations in MM-ST21pv). As a baseline for future work on biomedical concept recognition, both mention level and document level metrics for the TaggerOne model, computed on the MM-ST21pv Test subset, are reported in Table 7.

## 5. Conclusion

We presented the formal release of a new resource, named MedMentions, for biomedical concept recognition, with a large manually annotated annotated corpus of over 4,000 abstracts targeting a very large fine-grained concept ontology consisting of over 3 million concepts. We also included in this release a targeted sub-corpus (MedMentions ST21pv), with standard training, development and test splits of the data, and the metrics of a baseline concept recognition model trained on this subset, to allow researchers to compare the metrics of their concept recognition models.

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
