# OpenReview forum: "MedMentions: A Large Biomedical Corpus Annotated with UMLS Concepts"
_AKBC.ws/2019/Conference — AKBC 2019_

### Official Review · AnonReviewer1 · 2018-12-16
**Useful resource, but claims could be better supported, and the uniqueness of the resource better argued**

**Rating:** 6
**Confidence:** 5

**Review:**

The paper describes a new resource "Med Mentions" for entity linking of Pubmed abstracts, where entities are concepts in the UMLS type hierarchy -- for example "Medical Device".

The annotations were manually verified. (I assume the intention is to use this as a benchmark, but the paper does not say)

The paper is very rigorous in describing which concepts were considered, and which were pruned. Authors suggest to combine it with "TaggerOne" to obtain end-to-end entity recognition and linking system.

It is a little bit unclear what the main contribution of this paper is. Is it a benchmark for method development and evaluation (the paper mentions the train/dev/test split twice)? or do the authors propose a new system based on this benchmark?, or was the intent to test a range of baselines on this corpus (and what is the purpose?) -- I believe this lack of clarity could be easily addressed with a change in structure of headings. (Headings are currently not helping the reader, a more traditional paper outline would be helpful.)

I appreciate that the paper lists a range of related benchmarks. However, I am missing a discussion of: where the advantage of MedMentions is in contrast to these benchmarks? What is MedMentions offering that none of the other benchmarks couldn't?


It is indisputable that a new resource provides value to the community, and therefore should be disseminated. However, the paper quality is more reminiscent of a technical report. A lot of space is dedicated to supplemental information (e.g. page 6) which would be better spent on a clear argumentation and motivation of the steps taken.

---

> ### Author Response · Authors · 2019-01-17
> **Response to AnonReviewer1**
>
> The introduction has been updated to state that a new annotated gold standard resource ('benchmark') is being introduced.
>
> The main contribution in this paper is the new 'benchmark' - a manually annotated resource, for training and evaluating biomedical concept recognition systems. This new resource addresses 2 key needs:
> (i) Larger annotated resource, useful for training today's more complex ML models
> (ii) Broader coverage of biology and medicine concepts, by targeting UMLS.
>
> CRAFT is the closest, in size and coverage of biology. MedMentions can be viewed as a supplement -- the sizes of the ontologies is so much greater than the available annotated corpora that for ML models more data will always be useful. MedMentions has the added benefit of better coverage of concepts from some biomedical disciplines, e.g. diseases and drugs. This comes from the use of UMLS as the target ontology.
>
> TaggerOne is an established model for biomedical concept recognition and is offered here as a baseline concept recognition model, that researchers developing new models may compare their results against.
>
> Section 3 on related annotated corpora has been expanded slightly. Main differences are the size of the MedMentions benchmark corpus, and through the use of UMLS as the target ontology, more comprehensive coverage of biomedical concepts.

---

### Official Review · AnonReviewer3 · 2019-01-06
**Solid biomedical entity extraction/linking dataset**

**Rating:** 7
**Confidence:** 4

**Review:**

In this paper the authors introduce MedMentions, a new dataset of biomedical abstracts (PubMed) labeled with biomedical concepts/entities. The concepts some from the broad-coverage UMLS ontology, which contains ~3 million concepts. They also annotate a subset of the data with a filtered version of UMLS more suitable for document retrieval. The authors present data splits and results using an out-of-the-box baseline model (semi-Markov model TaggerOne (Leaman and Lu, 2016)) for end-to-end biomedical entity/concept recognition and linking using MedMentions.

The paper describes the data and its curation in great detail. The explicit comparison to related corpora is great. This dataset is substantially larger (hundreds of thousands of annotated mentions vs. ones of thousands) and covers a broader range of concepts (previous works are each limited to a subset of biomedical concepts) than previous manually annotated data resources. MedMentions seems like a high-quality dataset that will accelerate important research in biomedical document retrieval and information extraction.

Since one of the contributions is annotation that is supposed to help retrieval, it would be nice to include a baseline model that uses the data to do retrieval. Also, it looks like the baseline evaluation is only on the retrieval subset of the data. Why only evaluate on the subset and not the full dataset, if not doing retrieval?

This dataset appears to have been already been used in previous work (Murty et al., ACL 2018), but that work is not cited in this paper. That's fine -- I think the dataset deserves its own description paper, and the fact that the data have already been used in an ACL publication is a testament to the potential impact. But it seems like there should be some mention of that previous publication to resolve any confusion about whether it is indeed the same data.

Style/writing comments:
- Would be helpful to include more details in the introduction, in particular about your proposed model/metrics. I'd like to know by the end of the introduction, at a high level, what type of model and metrics you're proposing.
- replace "~" with a word (approximately, about, ...) in text
- Section 2.3: capitalization typo "IN MEDMENTIONS"
- Section 2.4, 3: "Table" should be capitalized in "Table 6"
- Use "and" rather than "/" in text
- Section 4: maybe just say "training" and "development" rather than "Training" and "Dev"
- 4.1: Markov should be capitalized: semi-Markov
- 4.1: reconsider use of scare quotes -- are they necessary? 'lexicons', 'Training', "dev', 'holdout'
- 4.1: replace "aka" with "i.e." or something else more formal. In general this section could use cleanup.
- 4.1: last paragraph (describing metrics, mention-level vs. document-level) is very confusing, please clarify, especially since you claim that a contribution of the paper is to propose these metrics. Is it the case that mention-level F1 is essentially entity recognition and document-level is entity linking? An example could possibly help here.

---

> ### Author Response · Authors · 2019-01-17
> **Response to AnonReviewer3**
>
> The Introduction has been expanded in the revised submission to make the motivation more explicit.
>
> In this paper we wanted to describe the new resource we have created for training and evaluating biomedical concept recognition in scientific literature. The resource addresses 2 key needs:
> (i) Larger annotated resource, useful for training today's more complex ML models
> (ii) Broader coverage of biology and medicine concepts, by targeting UMLS.
>
> Information Retrieval models is a research area on its own, and in this paper we just wanted to focus on concept recognition (CR). Metrics for CR models are quite standardized now, and the specific ones we use are described in a new section 4.1 added to the revised submission.
>
> Citation of the (Murty et al., ACL 2018) paper was excluded to anonymize the paper for review. It will be included in the final copy.
>
> Thanks for the detailed proof-reading. These should now be fixed in the uploaded revision.

---

### Official Review · AnonReviewer2 · 2019-01-07
**MedMentions: A Large Biomedical Corpus Annotated with UMLS Concepts**

**Rating:** 7
**Confidence:** 3

**Review:**

The paper “MedMentions: A Large Biomedical Corpus Annotated with UMLS Concepts” details the construction of a manually annotated dataset covering biomedical concepts. The novelty of this resource is its size in terms of abstracts and linked mentions as well as the size of the ontology applied (UMLS).
The manuscript is clearly written and easy to follow. Although other resources of this type already exist, the authors create a larger dataset covered by a larger ontology. Thus, allowing for the recognition of multiple medical entities at a greater scale than previously created datasets (e.g. CRAFT).
Despite the clarity, this manuscript can improve the following:
Section 2.3 – How many annotators were used?
Section 2.4, point 2 - The process used to determine biomedical relevance is not detailed. Section 4.1 - No reason is given for the choice of TaggerOne. In addition, other datasets could have been tested with TaggerOne for comparison with the MedMentions ST21pv results.
Misspelling and errors in section 2.3: “Rreviewers”, “IN MEDMENTIONS”
Overall, this paper fits the conference topics and provides a good contribution in the form of a large annotated biomedical resource.

---

> ### Author Response · Authors · 2019-01-17
> **Responses to  AnonReviewer2**
>
> More detailed information on the annotation process and inter-annotator agreement is being gathered and will be published on the release site.
>
> Section 2.4 point 2 has been expanded to describe "biomedical relevance" as used to select semantic types.
>
> The TaggerOne based model is now described in (expanded) section 4.2, and reason for its selection is also included. The TaggerOne paper referenced does include performance metrics on other biomedical datasets. Its demonstrated performance on recognizing biomedical entities from multiple types was our reason for using it as a baseline.  Our goal at present is to simply offer baseline metrics for concept recognition models trained on MedMentions ST21pv.
>
> Typos fixed, thanks!

---

### Author Response · Authors · 2019-01-17
**Revised submission**

We would like to thank the reviewers for their detailed review. A revised version of the paper has been uploaded that should address most of the points raised.  Various sections have been expanded in the revision, including: the introduction, to make the motivation more explicit; a clearer description of the metrics used for evaluating concept recognition models. See also responses to some specific questions below.

---

### Meta-Review · Area_Chair1 · 2019-02-11
**Good paper about a valuable new data set**

**Recommendation:** Accept (Poster)
**Confidence:** 5

**Metareview:**

The paper provides a valuable new resource to the community, a data set of 350,000 mentions from 4000 abstracts, all linked to UMLS concepts.  MedMentions has some advantages over existing datasets that are either smaller in size, narrower in coverage of concepts, or only provide weakly supervised labels of the mentions (i.e., concepts are associated with an abstract, but not explicitly identified as mentions therein). The reviewers all agree that MedMentions would be a valuable resource for the community.  The main criticism of the paper is that the motivation and contribution were not initially clear; however, the authors have addressed this criticism in the responses and have already updated the introduction to make the motivation and contribution more explicit.

---

### Decision · Program_Chairs · 2019-02-15
**AKBC 2019 Conference Decision**

Accept